# Effects of Forest Healing Anti-Aging Program on Psychological, Physiological, and Physical Health of Older People with Mild Cognitive Impairment

**DOI:** 10.3390/ijerph19084863

**Published:** 2022-04-16

**Authors:** Ji-Eun Baek, Jin-Hwa Jung, Ho-Jin Shin, Sung-Hyeon Kim, Si-Yoon Sung, Su-Jin Park, Suk-Chan Hahm, Hwi-Young Cho, Min-Goo Lee

**Affiliations:** 1Department of Health Science, Gachon University Graduate School, Incheon 21936, Korea; baekjieun421@gmail.com (J.-E.B.); gpgkorea89@gmail.com (H.-J.S.); gpgkorea30@gmail.com (S.-H.K.); 2Department of Occupational Therapy, Semyung University, Jecheon 27136, Korea; otsalt@nate.com; 3Huenlab, Seoul 05542, Korea; cecilq@naver.com; 4Forest Policy and Economics Department, Forest Welfare Division, National Institute of Forest Science, Seoul 02455, Korea; snowshoe@korea.kr; 5Graduate School of Integrative Medicine, CHA University, Seongnam 13488, Korea; schahm@cha.ac.kr; 6Department of Physical Therapy, Gachon University, Incheon 21936, Korea; 7Department of Physiology, Korea University College of Medicine, Seoul 02841, Korea

**Keywords:** older people, forest therapy, mild cognitive impairment, physical health, physiological health, psychological health

## Abstract

This study aimed to determine the effect of a forest healing anti-aging program on psychological, physiological, and physical health in older people with mild cognitive impairment (MCI). Twenty-two older people with MCI living in the city participated in a forest healing anti-aging program. Psychological indicators included the mini-mental state examination (MMSE), Beck depression inventory (BDI), profile of mood states (POMS), World Health Organization Quality of Life instrument (WHOQOL), and the Pittsburgh sleep quality index (PSQI). Physiological indicators included vital signs, body composition, and blood analysis. Physical indicators included the senior fitness test (SFT), muscle strength, spatiotemporal parameter of gait, static balance, and dynamic balance. Psychological, physiological, and physical indicators were evaluated at first and second pre-measurement, post-measurement, and one-month follow-up. MMSE, BDI, POMS, WHOQOL, body composition, blood analysis, SFT, muscle strength, spatiotemporal parameter of gait, and dynamic balance were significantly different between pre- and post-measurement. Beck depression inventory, POMS, WHOQOL, PSQI, SFT, muscle strength (elbow flexor muscle, knee extensor muscle), spatiotemporal parameter of gait significantly improved continually until the one-month follow-up. In conclusion, the forest healing program had a positive effect on the psychological, physiological, and physical health of older people with MCI.

## 1. Introduction

Modern society is facing a population aging phase due to advances in medical technology and a low birth rate. According to the National Statistical Office of South Korea, the average life expectancy worldwide as of 2020 is 72.3 years, and by 2100, 80 years later, it is expected have increased by approximately 10 years to 81.7 years. In 2018, the proportion of the population aged 65 and over reached 14.3% in Korea, entering an aging society, which is expected to become a post-aged society by 2025 [1]. The aging phenomenon is not only occurring in Korea but also around the world, which means that the social economy of future societies will be very different from the present.

Dementia is one of the main causes of disability and dependence in older people around the world. Cognitive function of older people with dementia is significantly lower than that of normal older people of the same age, which negatively affects overall brain activities such as memory, thinking, understanding, learning, language judgment, and calculation. In 2015, the global socioeconomic cost of dementia was estimated to be $800 billion, equivalent to 1.1% of global gross domestic product (GDP) [2]. One way to mediate this rapid increase in the prevalence of dementia is to detect and cope with mild cognitive impairment (MCI) early. MCI refers to an intermediate stage between the decline in expected cognitive function due to normal aging and severe cognitive decline due to dementia [3]. The importance of treatment in older people with MCI is easily overlooked because their memory is slightly decreased, but their ability to perform daily activities is maintained. However, the conversion rate of dementia in people over 65 years of age who have MCI is approximately 15%, which is more than five times higher than that of older people without MCI. In Korea, the number of MCI patients reached 230,000 in 2018, an increase of approximately ten times compared to 2010, and the total cost of treatment reached 30.3 billion dollars in 2014. As the age of onset is also decreasing gradually, early detection and management of MCI, which is highly likely to progress to dementia, are necessary [4].

In an aging society, it is important not only to prolong life but also to lead a healthy life through well-being. However, in many countries, serious environmental pollution has occured due to rapid urbanization along with the aging population. This causes various psychological stresses or lowers the quality of life in older people [5]. Various intervention methods have been proposed worldwide to improve the physical and psychological health of older people, and among them, interventions through nature are emerging. Previous studies have reported that greater exposure to nature has a positive effect on human cognition and emotion [6], and physical activity in an environment such as a forest or grassland has a much more positive effect on mental health than indoor exercise [7,8]. In addition, it has been found that exercise that involves different landforms and altitudes, such as trekking, has positive effects on increasing the amount of physical activity [9]. In particular, in countries with abundant forest resources, forest healing is being proposed as a therapeutic intervention method for older people and various diseases.

Forest therapy is a natural therapy defined as activities that improve the health of the human body by utilizing various physical and environmental factors of nature. Its main purpose is to help maintain health and enhance immunity [10]. In Korea, 63.4% of the land is forest, and the Korea Forest Service is developing and distributing several forest healing programs using forest resources in response to the aging population and increase in chronic diseases. The Korea Forest Welfare Institute, an organization affiliated with the Korea Forest Service, operates forest healing programs at 15 forest welfare facilities, including the National Center for Forest Therapy, to provide optimal forest welfare services to the public [11].

To date, many positive effects of forest healing have been reported on the human body. People who have undergone forest therapy have decreased depression [12] and cortisol levels [13], and their mood states have improved [14]. It also reduces blood pressure [15] and activates parasympathetic nerves [16]. In a study that implemented a forest healing program for older people, it was reported that the cardio-ankle vascular index and forced expiratory volume were improved [17] and melatonin levels increased [18]. However, no studies have reported the effects of forest therapy on older people with MCI. Therefore, this study aimed to verify the effect of the forest healing anti-aging program on the psychological, physiological, and physical health of older people with MCI.

## 2. Materials and Methods

### 2.1. Participants

The participants of this study were recruited through poster advertisements at the Seoul Metropolitan Center for Dementia (37.47, 126.90) located in Geumcheon-gu, Seoul, from July to August 2020.

Among those wishing to participate, we recruited those who were 60 years or older, able to walk 10 m independently without a walking aid and had a score between 23 and 27 in the Korean version of the Mini-Mental State Examination (MMSE-K) [19]. The exclusion criteria were as follows: (1) persons who lack Korean skills and have limited communication, (2) persons with Parkinsonism or motor-impairing neurological disorders (e.g., stroke), (3) persons whose musculoskeletal disorders in the lower extremities (e.g., severe osteoarthritis or a history of knee/hip joint replacement surgery) could affect the performance of walking during a clinical examination, (4) persons who have taken neuroleptics or benzodiazepines and major depression drugs, (5) persons whose Beck depression index are 19 or more points, (6) persons who have visited the forest at least once a week, and (7) persons who perform moderate-intensity aerobic physical activities for more than 300 min per week.

To calculate the sample size, G*Power 3.1.9.7 was used, with an effect size of 0.25, a significance level of 0.05, and an α power of 0.8. Based on these values, with a 20% dropout rate, a total of 27 subjects were required. Therefore, 22 people participated in the experiment, and all participants completed the research participation agreement form before participating in the study. This study was approved by the Bioethics Review Committee of the Gachon University (1044396-202006-HR-111-02) and was registered with the Clinical Research Information Service (CRIS; KCT0005365).

### 2.2. Study Sites

This study was conducted at the National Center for Forest Therapy, located in Yecheon-gun (36.83, 128.47) and Yeongju-si (36.87, 128.49), Gyeongsangbuk-do, South Korea. The National Center for Forest Therapy consists of a central facility district of about 142 ha with buildings such as Forest of Healing, Barefoot Garden, and Forest Healing Camp Center. The forest path around the National Center for Forest Therapy is connected to Sobaeksan National Park, Myojeokbong, and Cheonbusan, which is approximately 50 km long and has a section with a deck road with an inclination of less than 8%, so that even older people can hike without difficulty.

The average temperature in September–October at the location was 21.43–11.60 degrees Celsius, and in fall, the five components (α-Pinene, β-Pinene, Camphene, Limonene, Camphor) with the highest concentration among the natural volatile organic compounds (NVOCs) were present at a rate of 77.91 pptv [20].

In this area, tree species, such as *Pinus koraiensis Siebold* and *Zucc.*, *Pinus densiflora Siebold and Zucc.*, and *Quercus mongolica Fisch. ex Ledeb.* form a mixed forest. In addition, various plant groups such as *Larix kaempferi (Lamb.) Carrière*, *Lindera obtusiloba Blume*, *Koelreuteria paniculata Laxm.*, and *Rhododendron mucronulatum Turcz.* grow wild.

### 2.3. Procedure

The recruitment of participants for this study started in July and was completed in August 2020. The first pre-measurement started in mid-August 2020, and the intervention and evaluation were completed in November 2020. The flowchart in Figure 1 shows the research procedure that was used in this study.

Before participating in the study, the participants were asked for their consent to use their personal information and were informed that all data would be encrypted and processed. Those who agreed to participate signed a consent form, and all research procedures were conducted in compliance with the Declaration of Helsinki of the World Medical Association.

Before the start of the forest healing anti-aging program, information on general characteristics and psychological, physiological, and physical indicators for 22 participants were evaluated using two pre-measurements. The participants performed two pre-measurements before participating in the program. To prove that a participant’s condition was chronic, the second pre-measurement was performed at an interval of at least one month after the first pre-measurement. On the day of the second pre-measurement, all participants underwent a forest healing anti-aging program for two days and one night over three weeks. After the schedule was completed, the measurement items that were performed during the two pre-measurements were evaluated in the same way to verify the effectiveness of the program. All evaluations were performed by one medical doctor, one nurse, and four physical therapists with more than three years of clinical experience.

### 2.4. Measurement

Except for blood analysis, all variables were evaluated at the first and second pre-measurement, post-measurement 4 h after the last intervention, and one-month follow-up-measurement. Blood collection and analysis were performed at the second pre-measurement and post-measurement.

#### 2.4.1. Psychological Variables

The MMSE-K was used to screen the participants’ cognitive status and examine the changes in cognitive function. The MMSE-K has a total of 30 points, of which mild cognitive impairment falls between 23 and 27 points. This tool has high sensitivity and specificity and is widely used because of its excellent validity (0.93) [21].

The Korean version of the Beck depression inventory (K-BDI) was used for participant screening and depressive status. The K-BDI is a self-reporting tool designed to evaluate the presence and severity of depression. It consists of 21 items covering the cognitive, emotional, motivational, and physical domains of depression, and each item is scored on a scale of 0 to 3, for a total of 63 points. A score of 0–9 means no or minimal depression, a score of 10–18 means little or moderate depression, a score of 19–29 means moderate to severe depression, and a score of 30–63 means very severe depression. The overall reliability coefficient of the BDI is 0.882, and the test-retest reliability is 0.93, which is very high [22].

The Korean version of the profile of mood states (K-POMS) was used to examine the participants’ mood states. This tool evaluates an individual’s current feelings and attitudes using a five-point scale across 65 items. The mood state scale consists of six items: tension-anxiety (TA), depression-dejection (DD), anger-hostility (AH), fatigue-inertia (FI), confusion-bewilderment (CB), and vigor-activity (VA). After summing up the corresponding scores, the score of the VA item is excluded to calculate the total mood disturbance (TMD) score. Currently, the higher the total score in the remaining areas (except for the VA item), the more severe the mood disorder. The overall reliability coefficient of the K-POMS is 0.93 [23].

The Korean short version of the WHOQOL (K-WHOQOL-BREF) was used to measure the participants’ quality of life. This tool consists of 26 questions in five areas: overall quality of life and general health (OG), physical health (PH), psychological, social relationship (SR), and environmental health, and each question can be answered on a five-point Likert scale. The higher the score in each domain, the more positive the evaluation of the quality of life. The test-retest reliability of this scale is 0.841 and the overall reliability coefficient is 0.898 [24].

The Korean version of the Pittsburgh sleep quality index (K-PSQI) was used to evaluate the sleep quality of the participants. This tool is self-reported and consists of seven areas and 18 items. The details are as follows: (1) sleep quality, (2) sleep latency, (3) sleep duration, (4) habitual sleep efficiency, (5) sleep disturbance, (6) use of sleeping medication, and (7) daytime dysfunction. The sleep index can be calculated by adding up the scores of all seven items, out of a total of 21 points. The higher the score, the lower the quality of sleep. The overall reliability coefficient of this scale is 0.84 and the test-retest reliability is 0.65 [25].

#### 2.4.2. Physiological Variables

Blood pressure and heart rate were measured using an automatic electronic sphygmomanometer (ACCUNIQ BP203, SELVAS Healthcare, Daejeon, Korea) to evaluate vascular risk factors, which are risk factors for dementia. The participants were placed on a curve with one arm relaxed and constant breathing.

A body composition analyzer (Inbody470, InBody Co., Seoul, Korea) was used to examine body composition changes in the participants. The tested variables were skeletal muscle mass, body fat mass, and total body water. The participants stood barefoot on the device and took correct measurement postures according to the evaluator’s instructions.

Approximately 5 mL of blood was collected by injection through a vein to examine biomarkers indicating the health status, stress, and emotional level of the participants. Blood was collected at 8 a.m. for the second pre-measurement and post-measurement by direct injection by medical personnel with more than 10 years of experience. The factors identified in the collected blood were as follows: cathepsin B, insulin-like growth factor 1 (IGF-1), cortisol, and C-reactive protein (CRP). Cathepsin B has been reported to be related to long-term memory [26] and IGF-1 has been reported to be related to the response time to stimuli [27]. Cortisol is a major factor related to stress [28], and it has been reported that CRP is related to chronic stress [29]. The collected blood was assigned a code for each participant so that personal information could not be identified. Blood analysis was performed by SCL Healthcare (Yongin-si, South Korea), an external analysis company. The reagents and analytical equipment used for each factor analysis are listed in Appendix A (Table A1).

#### 2.4.3. Physical Variables

Functional fitness was assessed using the senior fitness test (SFT). The six components of the SFT are as follows: (1) arm curl, (2) chair stand, (3) 2-min step walk, (4) back scratch, (5) chair sit and reach, and (6) 8-ft up and go. This tool allows the evaluation of strength, endurance, flexibility, dynamic balance, and agility of the upper and lower extremities [30]. When the SFT was performed on older people with cognitive impairment, the test-retest reliability ranged from 0.93 to 0.98 [31].

The muscle strength of the elbow flexor (EF), knee extensor (KE), and ankle plantar flexor (APF) were measured using a digital hand-held dynamometer (microFET^®^, Hoggan Scientific, Salt Lake City, UT, USA), and a digital dynamometer (Jamar Plus, Sammons Preston, Bolingbrook, IL, USA; Jamar Pinch Gauge, Patterson Medical, Bolingbrook, IL, USA) was used to assess handgrip strength and finger pinch strength. The participant performed elbow flexion and knee extension according to the evaluator’s instructions in a sitting position in a chair and ankle plantar flexion in the supine position, while measuring the pushing force by placing the portable dynamometer in the opposite direction of the motion. The digital dynamometer evaluated hand and finger pinch strength using gripping tools in a sitting position on a chair. All tools used to measure muscle strength have a very high level of reliability, above 0.90 [32,33].

A GAITRite walkway (GAITRite^®^, CIR-systems Inc., Sparta, NJ, USA) with a size of 457.2 cm × 90.2 cm × 64 cm was used to analyze spatiotemporal changes in walking. To exclude accelerators and decelerators from the measurements, participants were instructed to start walking two meters in front of the mat’s start point and walk at a self-selected comfortable pace up to a marked line two meters behind the mat’s endpoint. Measurements were made twice, with a one-minute rest period between measurements. The sampling rate was set to 100 Hz, and data were collected and analyzed using GAITRite Platinum software (version 4.7.7), a dedicated software for GAITRite^®^. The measurements collected were as follows: velocity (cm/s), cadence (steps/min), step length (cm), stride length (cm), step time (s), swing time (s), stance time (s), and cycle time (s). The reliability of this tool is 0.92, which is very high, and the intra-rater correlation coefficient is 0.96 or more [34].

To evaluate static balance ability, an AMTI AccuSway (Advanced Mechanical Technology, Inc., Watertown, MA, USA) was used, and the participant’s sway velocity and sway area were analyzed in the standing position with eyes open and eyes closed. The reliability of this tool is 0.70–0.89 [35]. Dynamic balance ability was evaluated using the modified functional reach test by measuring the distance to reach forward, left, and right in a sitting position in a chair. The starting point was marked by raising the arm so that the participant’s trunk and upper arm formed a 90° angle in the direction to be measured. The intra-rater reliability of the measurement method is 0.96 or higher [36].

### 2.5. Intervention

Participants performed one session per week, two days per session, for a three-week forest healing program at the National Center for Forest Therapy. The forest healing program was based on the forest healing anti-aging program developed by the Korea Forest Service, which consists of detailed items that evenly cover the areas of the body, cognition, emotion, and nutrition (Table 1 and Table 2).

During the intervention period of two days and one night, the participants stayed only at the facility within the National Center for Forest Therapy, and lodging and meals were also used within the facility. To move between programs, walking was used.

All interventions were conducted under the guidance of five forest healing instructors from the National Center for Forest Therapy who had extensive experience in guiding forest healing programs. The forest healing instructors who conducted the intervention had a bachelor’s degree or higher and had at least one year of experience in forest healing.

### 2.6. Statistics Analysis

All statistical analyses were performed using SPSS software version 25.0 (SPSS Inc., Chicago, IL, USA). Normality testing was performed using the Shapiro–Wilk test. A paired *t*-test was used for variables that satisfied normality to examine the difference between the first and second pre-measurements, and the Wilcoxon signed-rank test was used for variables that did not satisfy normality. A repeated measures analysis of variance (ANOVA) and paired *t*-test were used for variables satisfying normality to compare differences in measured values at each time point, and the Friedman test and the Wilcoxon signed-rank test were used for variables that did not satisfy normality. Post hoc analysis using the Wilcoxon signed-rank test was performed applying Bonferroni correction, and as a result, the significance level was set to *p* < 0.017. The statistical significance level was set at α = 0.05.

## 3. Results

### 3.1. General Characteristics of Participants

Twenty-six older city dwellers with MCI who met the inclusion criteria were recruited and participated in the first pre-measurement. Three participants later dropped out before the second pre-measurement due to personal circumstances, and additional participants dropped out after one intervention; thus, a total of 22 people finally participated in the study. The descriptive statistics of the participants’ general characteristics, such as age, sex, and body measurements, are shown in Table 3.

### 3.2. Baseline Background Characteristics

Among the variables measured at both time points, the difference between the first and second pre-measurements was not significant for all variables (*p* > 0.05) (Appendix A, Table A2). Since there was no significant difference between the first and second pre-measurements, the second pre-measurement values were set as the pre-measurement value (Pre) to investigate the effect of the forest healing anti-aging program. Blood analysis was only performed at the second pre-measurement.

#### 3.2.1. Psychological Effects of the Forest Healing Anti-Aging Program

There was a significant difference in the MMSE scores (*p* < 0.05). There were significant differences between the pre- and post-measurements and between the post- and follow-up measurements (*p* < 0.017).

The BDI score was significantly different (*p* < 0.05). There were significant differences between the pre- and post-measurements and between the pre- and follow-up measurements (*p* < 0.05).

Among the POMS subcategories, there were significant differences in DD, AH, FI, and TMD (*p* < 0.05). DD, FI, and TMD had a significant difference between the pre- and post-measurements, and the items with a significant difference between pre- and follow-up measurements were DD, AH, FI, and TMD (DD, AH, TMD: *p* < 0.017; FI: *p* < 0.05).

Among the subcategories of the WHOQOL-BREF, there were significant differences in OG, PH, and SR (*p* < 0.05). The item with a significant difference between pre- and post-measurements was SR, and the items with significant differences between the pre- and follow-up measurements were OG and PH (OG, SR: *p* < 0.017; PH: < 0.05).

In the total PSQI score, there was a significant difference between the pre- and follow-up measurements (*p* < 0.05). There were no significant differences in the sub-items of the PSQI (*p* > 0.05) (Table 4).

#### 3.2.2. Physiological Effects of the Forest Healing Anti-Aging Program

There was no significant difference in all variables of vital signs (*p* > 0.05).

The body composition showed significant differences in the pre- and post-measurements and in the post- and follow-up measurements in all variables (*p* < 0.05).

Blood analysis revealed a significant difference in the cortisol levels (*p* < 0.05) (Table 5).

#### 3.2.3. Physical Effects of the Forest Healing Anti-Aging Program

Among the sub-items of SFT, there were significant differences between the pre- and post-measurements and between the pre- and follow-up measurements in AC, CS, 2SW, and 8UG (*p* < 0.05).

Among the dynamometer results, there were significant differences in EF, KE, and APF (*p* < 0.05). The items with significant differences between pre- and post-measurements were EF, KE, and APF, and the items with significant differences between pre- and follow-up measurements were EF and KE (EF, KE, APF: *p* < 0.05).

Significant differences were observed in all spatiotemporal gait variables (*p* < 0.05). The post hoc analysis showed a significant difference between the pre- and post- measurements in all variables, and the items with a significant difference between the pre- and follow-up measurements were swing time and stance time. The items that were present were step length and stride length (velocity, step length, stride length, swing time: *p* < 0.05; cadence, step time, stance time, cycle time: *p* < 0.017).

In the modified FRT, there was a significant difference between the pre- and post-measurements in the right and left directions (*p* < 0.05) (Table 6).

## 4. Discussion

This study was conducted to investigate the effects of the forest healing anti-aging program on the psychological, physiological, and physical health of older people with MCI, and the following results were obtained. First, the forest healing anti-aging program improved the psychological health of older people with MCI. Second, the forest healing anti-aging program improved the physiological health of older people with MCI. Third, the forest healing anti-aging program improved the physical health of older people with MCI.

### 4.1. Effects of Forest Healing Anti-Aging Program on Psychological Health

A study by Yu and Hsieh showed that the cognitive ability of participants who participated in a three-day forest healing workshop improved by 27.74% [37]. Similarly, the forest healing program improved scores in measures of cognitive ability of older people with mild cognitive impairment by 5.1%. Moreover, as the scores of BDI and DD of POMS decreased significantly, it was found that the forest healing anti-aging program was effective in improving depression in older people. Depression in older people tends to increase due to a decrease in physical health that occurs with increasing age, which may lead to a decrease in social relationships and QOL [38]. The participants experienced a new forest environment where they could experience various sensory stimuli (visual, auditory, and proprioceptive senses) outside the city where they normally live. In addition, to solve the task, group activities were performed through interactions with various people, and insufficient social relationships improved. Through these activities, participants were able to relieve negative emotions and obtain emotional stability, which was shown to improve the POMS and WHOQOL.

In the case of the PSQI, there was no significant difference in the sub-items but there was a significant change in the total score. Similarly, in a previous study, there was no significant change in PSQI when forest healing was performed over five nights and six days for cancer patients, but an increase in sleep time and sleep efficiency was found in the polysomnograph for evaluating sleep records [39]. In a previous study, the time spent in the forest per session was longer than that in our study; therefore, it was easy for the participants to adapt to the environment, and it seems that a positive effect was shown in the sleep record, which is a more objective indicator. In addition, among the participants in this study, those who were taking sleeping pills expressed that they experienced an increase in overall sleep quality, for example, they said that the drug dose decreased, or they stopped taking the drug after the forest healing intervention. Therefore, it is necessary to investigate the effect of the program on psychological health through adjustment for the intervention period and composition in a follow-up study.

### 4.2. Effect of Forest Healing Anti-Aging Program on Physiological Health 

In terms of body composition, the skeletal muscle mass and total body water increased, and fat mass decreased. This seems to be because the amount of physical activity increased throughout the program. A decrease in physical activity in the older population decreases skeletal muscle mass and increases fat mass, thereby increasing the probability of sarcopenia and obesity [40]. A vicious cycle leading to physical deterioration can accelerate cognitive decline by lowering brain activity [41].

In factors related to muscle and stress in the blood, cortisol, a stress hormone, was significantly reduced, indicating that the forest healing program was effective in reducing the stress level of the participants. In previous studies, decreased cortisol levels were associated with greater exposure to forest environments, and older people experienced a decrease in cortisol levels after low-to-moderate exercise [42,43]. Although the participant’s baseline CRP level did not show a significant change due to the low value (1.07 mg/L) with a high standard deviation, CRP also showed a tendency to decrease. A positive effect of the anti-aging forest healing program can be expected through the tendency to decrease CRP, a plasma biomarker that is significantly related to chronic stress and cardiovascular disease. In the case of cathepsin B and IGF-1 related to cognitive function, there was no significant change, but there was a tendency to increase; therefore, a follow-up study is needed to investigate the effectiveness through the modification of the intervention period and intensity.

### 4.3. Effects of Forest Healing Anti-Aging Program on Physical Health

In this study, the forest healing anti-aging program significantly improved the functional fitness, muscle strength, gait function, and dynamic balance ability of the participants. Interestingly, to the best of our knowledge, no studies have investigated the effects of forest healing on physical health. This study is the first attempt to identify this, and there are limitations in comparing the effectiveness of this study with other studies.

The participants of this study were 60-year-olds who exercised for 300 min or less per week. The forest healing anti-aging program involved a physical activity-based task in the mountains for approximately 376 min for two days per session. Various inclinations and irregular surfaces, walking speeds, and physical activity with others would have improved the participants’ motor functions. In particular, the WHO recommends that moderate-intensity aerobic physical activity of 300 min or more per week has health benefits [44].

In this study, participants who experienced forest healing showed improvements in functional fitness by improving upper and lower extremity muscle strength, endurance, dynamic balance, and muscle strength of the elbow flexor, knee extensor, and ankle plantar flexor muscles. In addition, gait time, length, and velocity significantly improved, and dynamic balance in the sitting position increased. These results are presumed to induce improvement in various motor functions because the physical activity-based forest healing anti-aging program consists of various activities that require whole-body movement.

Kozakai et al. reported that an increase in leisure-time physical activity can improve muscle function in middle-aged and older populations [45]. Mountain hiking, which is a type of leisure, can have a great influence on proprioceptive sensory input because the slope, altitude, and ground conditions change every moment [46]. However, there was no significant improvement in some physical health factors (flexibility, grip strength, pinch grip strength, static balance, etc.). This is presumably because the program provided in this study consisted of movements that require gait and global muscle activity of the trunk and lower extremities rather than specific movements that require functional movements of the hands, flexibility, and static balance.

### 4.4. Sustainability of Forest Healing Program Effects

In psychological health, the follow-up effect was maintained in most variables, but in the MMSE, the effect was significantly reduced, and there was no difference from before the intervention (*p* > 0.017). In terms of physiological health, body composition, which showed a significant improvement in all variables immediately after the intervention, returned to the same level as that before the intervention (*p* > 0.017). Continuous activity is required to improve and maintain cognitive function and body composition in the older population with MCI. However, the improvement in most of the physical health indicators was found to remain effective after one month.

### 4.5. Significance and Limitations of This Study

Studies have demonstrated the psychological and physiological effects of forest healing applied to participants of various age groups, occupations, and diseases, but the identification of the physical effects has been insufficient. This study is valuable as it is the first study to simultaneously identify the psychological, physiological, and physical effects of applying a forest healing program to older people with MCI.

However, this study had several limitations. First, no control group could be compared with the forest healing program group in this study. As the intervention used in this study consists of an individual program that maximizes forest resources, it is difficult to apply the same intervention in an urban environment. Therefore, to explain the effect of this intervention, the setting of a control group for the same program, excluding only environmental factors, is required. Second, comparison with the urban forest environment was limited. This study was conducted in a facility designed to be suitable for the implementation of a forest healing program located in a high-altitude and forested environment, which would have provided the participants living in the city an opportunity to experience a completely new environment and activities. Therefore, it is necessary to verify whether this intervention has the same effect in an urban forest environment. Third, although 22 participants were targeted through sample size calculation, there is a limitation in generalizing the study results because of the small sample size. Fourth, blood analysis was performed only in the second pre-measurement and post-measurement. Therefore, it cannot be confirmed in this study whether the changes in blood analysis persist until one month later. Finally, this study did not clearly identify the effects of atmospheric, weather, and seasonal factors of forest healing on the psychological, physiological, and physical health of participants. In particular, the healthy effects of the different levels of biogenic volatile organic compounds (BVOCs) in the forest atmosphere were not investigated. As per previous studies, exposure to BVOCs was deemed to produce remarkable effects on human health [47]. Thus, these factors could be potential factors that can affect the effectiveness of forest healing, which we will elucidate in future studies.

## 5. Conclusions

The forest healing anti-aging program increased cognitive ability, decreased the degree of depression, and improved mood and sleep disturbances in older people with MCI. It also positively affected body composition and reduced stress hormones. Finally, the functional fitness and muscle strength, gait, and dynamic balance abilities of the participants improved.

## Figures and Tables

**Figure 1 ijerph-19-04863-f001:**
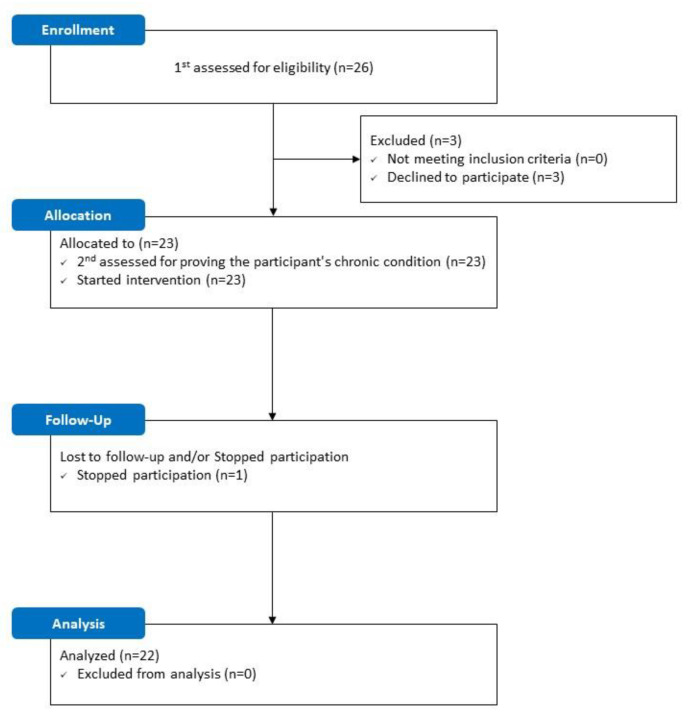
Flowchart.

**Table 1 ijerph-19-04863-t001:** The forest healing anti-aging program schedule.

Date	Time	Program	Place
1 day	12:00–13:00	Lunch	Cafeteria
	13:00–14:00	Orientation	Auditorium
	14:00–14:30	Warm-up for the brain and body	Forest
	14:30–15:00	Move to the place	
	15:00–16:00	Memorization training or Calculation training	Forest
	16:00–17:00	Recreational exercise	Forest
	17:00–18:00	Move to the place	
	18:00–19:00	Dinner	Cafeteria
	19:00–20:00	Eating meditation or Walking meditation	Wood cabin
	20:00–20:20	Sharing thoughts	Wood cabin
2 day	08:00–09:00	Breakfast	Cafeteria
	09:00–09:30	Warm-up for the brain and body	Forest
	09:30–10:00	Move to the place	
	10:00–11:00	Association training or Retrieval training	Forest
	11:00–12:00	Recreational exercise	Forest
	12:00–12:20	Sharing thoughts	Wood cabin
	12:20–	Lunch	Cafeteria

**Table 2 ijerph-19-04863-t002:** Composition of the forest healing anti-aging program.

Program	Description
Warm-up for the brain and body	Warm-up of light movements to activate the body and brain
Memorization training	Activities to observe and memorize kinds of natural objects while walking in the forest
Association training	Memorizing and associating milestones in the forest to create a story
Calculation training	Brain activation through mental arithmetic activities through understanding and memorizing tree types
Retrieval training	Activities to remember yesterday, today, and last week and express emotions through pictures or words
Recreational exercise	Stimulation of extremities of the body, promotion of blood circulation through stretching and strength exercise, improvement of exercise capacity
Eating meditation and tea meditation	Sensory activities focusing on movements and senses that occur when eating help stabilize the mind and body Eating nuts that are good for brain health (nutrition)
	Sensory activities focusing on movements and senses that occur when drinking tea help stabilize the mind and body Chrysanthemum tea to help calm the mind and body (nutrition)
Walking meditation	Focus on walking on forest paths, which stabilizes emotions and improves immunity
Sharing thoughts	Communicate and empathize with participants Mind and body stabilization through breathing meditation

**Table 3 ijerph-19-04863-t003:** General Characteristics of Participants.

Variables	Mean ± SD
Age (y)	61.90 ± 1.14
Male/Female (*n*)	7/15
Height (cm)	157.10 ± 7.18
Weight (kg)	59.54 ± 10.19
BMI (kg/m^2^)	24.02 ± 2.93
MMSE (score)	26.64 ± 0.79
BDI (score)	8.68 ± 5.69

Data are expressed as mean ± SD; BMI, body mass index; MMSE, mini-mental state examination; BDI, Beck depression inventory.

**Table 4 ijerph-19-04863-t004:** Effect of psychological variables.

Variables	Pre	Post	Follow-Up	*p*-Value	Post Hoc		
					Pre vs. Post	Pre vs. Follow-Up	Post vs. Follow-Up
MMSE (score)	26.64 ± 0.79	28.00 ± 1.27	27.14 ± 0.83	<0.001 ^†^	0.001 ^†^	0.022	0.001 ^†^
BDI (score)	8.36 ± 5.12	6.77 ± 4.47	6.14 ± 4.38	0.013 *	0.034 *	0.049^*^	0.777
POMS (score)							
Tension-anxiety	8.73 ± 4.10	7.09 ± 4.53	7.36 ± 3.98	0.208	-	-	-
Depression-dejection	12.27 ± 9.64	7.59 ± 7.32	7.64 ± 7.45	0.001 ^†^	0.002^†^	0.006 ^†^	0.896
Anger-hostility	7.00 ± 5.85	5.00 ± 5.62	4.41 ± 5.05	0.007 ^†^	0.086	0.001 ^†^	0.420
Fatigue-inertia	7.36 ± 4.93	5.59 ± 4.24	5.27 ± 4.03	0.001 *	0.021^*^	0.009 *	1.000
Confusion-bewilderment	8.23 ± 4.02	7.18 ± 3.39	7.32 ± 3.90	0.181	-	-	-
Vigor-activity	18.41 ± 6.08	19.59 ± 4.70	18.45 ± 5.02	0.058	-	-	-
Total mood disturbance	25.18 ± 24.86	12.86 ± 22.67	13.55 ± 21.79	0.001 ^†^	0.002 ^†^	0.003 ^†^	0.637
WHOQOL-BREF (score)							
Overall QOL and general health	6.45 ± 1.26	6.82 ± 1.37	6.95 ± 1.25	0.036 ^†^	0.059	0.012 ^†^	0.512
Physical health	23.18 ± 4.00	24.00 ± 3.65	24.77 ± 4.01	0.013 *	0.358	0.036 *	0.283
Psychological	18.23 ± 3.05	18.68 ± 3.50	19.36 ± 3.67	0.761	-	-	-
Social relationship	9.09 ± 1.15	10.00 ± 1.60	9.77 ± 1.57	0.007 ^†^	0.003 ^†^	0.030	0.364
Environmental	24.68 ± 4.70	25.00 ± 5.59	25.64 ± 5.18	0.447	-	-	-
Total	81.64 ± 12.45	84.50 ± 13.36	86.50 ± 14.11	0.400	-	-	-
PSQI (score)							
Sleep quality	1.23 ± 0.61	0.95 ± 0.58	1.09 ± 0.68	0.118	-	-	-
Sleep latency	1.45 ± 0.80	1.55 ± 1.06	1.36 ± 1.05	0.402	-	-	-
Sleep duration	0.41 ± 0.67	0.32 ± 0.65	0.18 ± 0.39	0.174	-	-	-
Habitual sleep efficiency	0.36 ± 0.85	0.09 ± 0.29	0.23 ± 0.69	0.074	-	-	-
Sleep disturbance	1.32 ± 0.57	1.18 ± 0.39	1.18 ± 0.39	0.325	-	-	-
Use of sleeping medication	0.23 ± 0.69	0.09 ± 0.43	0.00 ± 0.00	0.061	-	-	-
Daytime dysfunction	1.45 ± 0.74	1.45 ± 0.91	1.27 ± 0.94	0.328	-	-	-
Total	6.45 ± 3.04	5.64 ± 2.46	5.32 ± 2.15	0.022 *	0.076	0.047^*^	1.000

Data are expressed as mean ± SD; * significant difference at *p* < 0.05 in parametric statistical method; ^†^ significant difference in non-parametric statistical method; Pre, second pre-measurement; Post, post-intervention; Follow-Up, one-month follow-up measurement; MMSE, mini-mental state examination; BDI, Beck depression inventory; POMS, profile of mood state; PSQI, Pittsburgh sleep quality index.

**Table 5 ijerph-19-04863-t005:** Effect of physiological variables.

Variables	Pre	Post	Follow-Up	*p*-Value	Post Hoc		
					Pre vs. Post	Pre vs. Follow-Up	Post vs. Follow-Up
Vital signs							
Systolic pressure (mmHg)	128.55 ± 12.63	129.77 ± 14.05	124.14 ± 18.42	0.074			
Diastolic pressure (mmHg)	74.59 ± 7.59	77.23 ± 9.97	74.05 ± 11.86	0.072			
Pulse pressure (mmHg)	53.95 ± 8.39	52.55 ± 7.29	50.09 ± 8.27	0.070			
Heart rate (bpm)	73.55 ± 11.71	74.00 ± 7.93	74.91 ± 8.05	0.601			
Body compositions							
Skeletal muscle mass (kg)	23.15 ± 4.03	23.81 ± 4.44	23.17 ± 4.35	0.002 ^†^	0.001 ^†^	0.845	0.001 ^†^
Body fat mass (kg)	19.12 ± 3.61	16.88 ± 3.63	19.06 ± 3.77	<0.001 ^†^	<0.001 ^†^	0.709	<0.001 ^†^
Total body water (L)	30.55 ± 5.18	31.35 ± 5.68	30.59 ± 5.56	0.010 ^†^	0.003 ^†^	0.935	0.005 ^†^
Blood compositions							
Cathepsin-B (pg/mL)	183497.71 ± 71436.06	184544.84 ± 105302.59	-	0.189	-	-	-
Cortisol (μg/dL)	13.04 ± 3.31	7.01 ± 2.56	-	< 0.001 ^†^	-	-	-
IGF-1 (ng/mL)	121.40 ± 32.78	124.43 ± 29.00	-	0.390	-	-	-
CRP (mg/L)	1.07 ± 0.87	0.88 ± 0.74	-	0.073	-	-	-

Data are expressed as mean ± SD; ^†^ significant difference in non-parametric statistical method; Pre, second pre-measurement; Post, post-intervention; Follow-Up, one-month follow-up measurement.

**Table 6 ijerph-19-04863-t006:** Effect of physical variables.

Variables	Pre	Post	Follow-Up	*p*-Value	Post Hoc		
					Pre vs. Post	Pre vs. Follow-Up	Post vs. Follow-Up
Senior Fitness Test					-	-	-
Arm curl (*n*)	17.68 ± 4.06	24.68 ± 4.89	25.14 ± 5.06	<0.001 *	<0.001 *	<0.001 *	0.874
30-sec chair stand (*n*)	11.73 ± 3.71	16.95 ± 3.58	17.55 ± 3.35	<0.001 *	<0.001 *	<0.001 *	0.290
2-min step walk (*n*)	92.50 ± 12.63	111.95 ± 12.89	114.00 ± 12.59	<0.001 *	<0.001 *	<0.001 *	1.000
8-ft up and go (sec)	7.71 ± 1.13	6.64 ± 0.88	6.57 ± 0.73	<0.001 ^†^	<0.001 ^†^	<0.001^†^	0.548
Back scratch (cm)	−8.50 ± 10.81	−8.05 ± 11.00	−7.32 ± 10.77	0.123			
Chair sit and reach (cm)	13.09 ± 6.41	13.59 ± 9.29	15.27 ± 8.42	0.354			
Muscle strength (kg)							
Elbow flexor (kg)	11.35 ± 3.61	15.60 ± 4.83	16.61 ± 4.75	<0.001 *	<0.001 *	<0.001 *	0.433
Knee extensor (kg)	13.01 ± 1.85	14.21 ± 2.24	15.41 ± 2.48	<0.001 *	0.010	<0.001 *	0.114
Ankle plantar flexor (kg)	13.16 ± 2.08	14.14 ± 1.62	13.50 ± 2.13	0.042 *	0.014	1.000	0.285
Grip (kg)	25.52 ± 7.18	26.22 ± 6.96	26.45 ± 7.13	0.151			
Pinch (kg)	7.12 ± 1.46	7.34 ± 1.51	7.35 ± 1.70	0.409			
Spatiotemporal parameter of gait							
Velocity (cm/s)	121.02 ± 16.19	128.96 ± 15.19	127.04 ± 16.21	0.002 *	0.001 *	0.099	1.000
Cadence (steps/min)	119.99 ± 10.08	124.33 ± 9.95	123.71 ± 10.81	0.013 ^†^	0.001 ^†^	0.035	0.144
Step length—Rt. (cm)	61.03 ± 5.14	63.47 ± 4.95	61.39 ± 5.48	0.002 *	0.002 *	1.000	0.007^*^
Stride length—Rt. (cm)	121.06 ± 10.29	126.65 ± 9.65	123.39 ± 10.75	0.001 *	0.001 *	0.566	0.031 *
Step time—Rt. (sec)	0.50 ± 0.05	0.48 ± 0.04	0.49 ± 0.04	0.001 ^†^	< 0.001 ^†^	0.054	0.205
Swing time—Rt. (sec)	0.38 ± 0.03	0.37 ± 0.03	0.37 ± 0.03	0.001 *	0.004 *	0.030 *	1.000
Stance time—Rt. (sec)	0.62 ± 0.06	0.60 ± 0.06	0.60 ± 0.06	0.028 ^†^	0.004 ^†^	0.014 ^†^	0.673
Cycle time—Rt. (sec)	1.01 ± 0.09	0.97 ± 0.08	0.97 ± 0.08	0.025 ^†^	0.001 ^†^	0.049	0.217
Static balance							
EO—Sway area (mm^2^)	1.47 ± 0.87	1.43 ± 0.99	1.45 ± 0.69	0.244			
EO—Sway velocity (mm/s)	1.11 ± 0.29	1.14 ± 0.32	1.11 ± 0.18	0.811			
EC—Sway area (mm^2^)	3.13 ± 3.43	2.64 ± 2.25	3.13 ± 3.42	0.727			
EC—Sway velocity (mm/s)	1.80 ± 0.78	1.58 ± 0.57	1.75 ± 0.74	0.428			
Dynamic balance (cm)							
Anterior	49.41 ± 12.18	48.18 ± 8.27	51.82 ± 6.88	0.117			
Right	32.25 ± 3.07	34.68 ± 3.72	34.45 ± 5.63	0.049 *	0.012	0.265	1.000
Left	30.55 ± 5.46	33.41 ± 4.10	31.86 ± 5.63	0.026 *	0.015	0.816	0.344

Data are expressed as mean ± SD; * significant difference at *p* < 0.05 in parametric statistical method; ^†^ significant difference in non-parametric statistical method; Pre, second pre-measurement; Post, post-intervention; Follow-Up, one-month follow-up measurement; Rt., right side; EO, eyes open; EC, eyes closed.

## Data Availability

Not applicable.

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
