# Peer review of "Effects of Forest Healing Anti-Aging Program on Psychological, Physiological, and Physical Health of Older People with Mild Cognitive Impairment"

_ijerph, 2022, doi:10.3390/ijerph19084863_

Round 1

Reviewer 1 Report

This paper aimed to see the effectiveness of the forest therapy program towards elderly with MCI. The topic is of high interest of the research and practice communities. It is novel and timely. Unfortunately, it has many methodological issues it cannot be published in this present form. 

  • The literature review does not provide enough ground for the research question. Why does forest healing work? What are the underlying mechanisms? What are the gaps? What are the risks of not knowing? 'Nobody has done it before' means novelty of the research. However, it does not mean the research questions are necessary. Please provide a convincing argument why this paper is needed.
  • The method is what I had problem with the most. Thus far, the research design provides that the forest program at this place helps elderly people with MCI healthier. However, since there is no control and no other forest therapy programs for comparison, the generalizability of the research is questionable. The authors mentioned the issue within the discussion and stated that it was difficult to compare the activities in the urban environments, but the results could be compared it with other anti-aging programs for urban environments, or ones that did not participate in any program, for example. 
  • The results are well-reported. 
  • The discussion section should provide more implications for practitioners, researchers, and designers. However, currently because of the research design, it is fairly difficult to do so.  

I highly recommend resubmission after further examination of how control groups can be applied, additional data are collected, and the research questions are more refined. Good luck in your future research endeavor! 

Author Response

#REVIEWER 1

This paper aimed to see the effectiveness of the forest therapy program towards elderly with MCI. The topic is of high interest of the research and practice communities. It is novel and timely. Unfortunately, it has many methodological issues it cannot be published in this present form.

  • All authors would like to express our sincere thanks to the reviewers who appreciated the topic of this study. As you pointed out, the design of this study without the control group may seem inadequate to test the effectiveness of the intervention. To solve this problem, we performed two pre-measurements and confirmed the chronic condition of the subject. It was judged that our description may cause misunderstandings to reviewers and readers, and we have made many changes to the content of the manuscript in consideration of this.
  • We sincerely hope that our amendments and the attached Appendix will aid your understanding and also clear up any misunderstandings about the design of this study.

  1. The literature review does not provide enough ground for the research question. Why does forest healing work? What are the underlying mechanisms? What are the gaps? What are the risks of not knowing? 'Nobody has done it before' means novelty of the research. However, it does not mean the research questions are necessary. Please provide a convincing argument why this paper is needed.
  • Response:
  • First of all, all authors agree with your good point. We did not provide sufficient evidence for providing enough ground associated with the research question and reasons to suggest the necessity of the research topic. The necessity of this study is as follows, and the content of the manuscript has been modified to properly describe it.
  • Damage caused by environmental pollution (noise, air pollution, light pollution, etc.) that occurs in the process of increasingly serious urbanization is a major factor that lowers people's health and quality of life [1]. In order to improve and promote people's physical and mental health, various therapeutic intervention methods have been proposed around the world. According to many studies and literature, among these intervention methods, nature is suggested as a good approach that is safe, eco-friendly, and has few side effects [2]. Especially forest healing, which improves human health by using various forest resources, has been implemented in several countries, and its effectiveness is gradually being verified [3-5].
  • However, most of the previous studies presented results limited to the effects of forest healing on the psychological (mood state, quality of life, etc.) and physiological (blood pressure and stress factors, etc.) changes of the human body. When a person's psychological and physiological changes have a positive effect on the body, it will appear as an improvement in functional movement in order to perform well in the individual's social health and daily life. Therefore, it is a very necessary topic to identify the effect of forest healing on changes in basic and functional physical functions of humans in relation to the results of previous studies. As such, this study was conducted to prove the effect of forest healing from other aspects.
  • In the Republic of Korea, along with the social aging problem, geriatric diseases are being raised as a big problem. We wanted to provide older people with a less reluctant and easy access method for therapeutic intervention. Due to the nature of Korea's topography, forest resources are abundant, and many people use them for leisure and other purposes. We thought that these resources could improve the physical, physiological, and psychological health as well as the cognitive function of older people with mild cognitive impairment. To prove this, we designed and performed this study.
  • All authors have additionally described the following content in the Introduction section, and also modified the content of the main manuscript and revised references based on your good comments.

  • P2 line 65-78: In an aging society, it is important not only to prolong life but also to lead a healthy life through well-being. However, in many countries, serious environmental pollution has been caused due to rapid urbanization along with an aging population. This causes various psychological stresses or lowers the quality of life in older people [5]. Various intervention methods have been proposed worldwide to improve the physical and psychological health of older people, and among them, interventions through nature are emerging. Previous studies have reported that greater exposure to nature has a positive effect on human cognition and emotion [6], and physical activity in an environment such as a forest or grassland has a much more positive effect on mental health than indoor exercise [7-8]. In addition, it has been found that exercise that can experience different types of landforms and altitudes, such as trekking, has positive effects on increasing the amount of physical activity [9]. In particular, in countries with abundant forest resources, forest healing using this is being proposed as a therapeutic intervention method for older people and various diseases.

  1. The method is what I had problem with the most. Thus far, the research design provides that the forest program at this place helps elderly people with MCI healthier. However, since there is no control and no other forest therapy programs for comparison, the generalizability of the research is questionable. The authors mentioned the issue within the discussion and stated that it was difficult to compare the activities in the urban environments, but the results could be compared it with other anti-aging programs for urban environments, or ones that did not participate in any program, for example.
  • Response:
  • We respect your comments and thank you for pointing out the shortcomings of this study.
  • As you pointed out, it would have been better if we conducted the study as a randomized controlled trial. However, the world was in a situation where everything was changing rapidly due to COVID-19, and we also had to conduct research in consideration of the individual circumstances of the subjects and changes in the Korean government's policies. All authors did not know whether the subjects participated or not until just before the intervention, and the number of subjects finally recruited was insufficient to divide into two groups in a situation where the government's policy was changing every day. Also, the institution that supported the research fund wanted to focus on identifying the forest healing effect.
  • Although we designed the study as a single group without a control group, we conducted two pre-measurements at intervals of more than one month to sufficiently prove that the subject's condition was chronic as suggested in the Appendix. This method is already being used in some studies, and two other reviewers also found it very interesting.
  • We also suggested in the Discussion that it is difficult to compare the effectiveness of this program with other programs (lines 399-400).
  • Obviously, as you pointed out, I'm sure the paper will be of better quality if we further experiment with the control group. But in reality, it is very difficult to do it again. We described what you pointed out as a study limitation, and we also ask that you understand our situation.

  1. The discussion section should provide more implications for practitioners, researchers, and designers. However, currently because of the research design, it is fairly difficult to do so.
  • Response:
  • This study is the first experiment to find out what effect the forest healing anti-aging program has on the psychological, physiological, and physical health of the elderly with mild cognitive impairment. Due to forest therapy as an intervention, it is very difficult to design the same experiment that completely excludes natural factors, so many studies have found the effect through a single group [6-9]. However, since we also knew the weakness of the single group study, we proved that the subject's condition was chronic through two pre-measurements, and quantitatively revealed the results of this study by comparing it with studies with similar study designs as much as possible. You recommended us further experiments for the study, but this is almost impossible under the present circumstances. This is because the pandemic has reached a more serious situation than when we conducted the experiment.
  • As the situation improves, we will consider your valuable advice and conduct further experiments. All authors urge you to reconsider the value of your current research. Once again, we would like to express our deepest gratitude to you for analyzing and pointing out this study.

Reviewer 2 Report

A complex and prudent study, I liked the study design with a 1st and then a 2nd pre-measurement for verifying the chronic state, and then a post-intervention measurement and a 1-month follow-up, which are reasonable and valuable. The various aspects investigated simultaneously are also a strength of the work. Yet I have some questions and remarks, and there are a couple of points to clarify and explain more deeply in the article before finalizing it, especially regarding the pre-measurements and the description of the intervention.

  • I did not check it in details but references [10] and [12] may be mixed up.
  • The authors wished to investigate the “elderly”, and in reference [1] (and in general as well) the elderly population is aged 65 years and over. Still, “60 years or older” participants were recruited with a mean age 61.90 +/- 1.14 years, who are younger. Why?
  • The exclusion criteria are thorough and well, especially with excluding persons with regular (weekly) forest visits. Still, the last point in line 100, is difficult to interpret: “The exclusion criteria were as follows: … and (7) should perform moderate-intensity aerobic physical activities for more than 300 min.”
  • The performance and timing of a first AND second pre-measurement are quite obscure throughout the article until explanation in the Results section, in lines 272 and 273. Please explain this (why and when the 2 measurements were accomplished, and how close the 2nd pre-measurement was to the intervention) already in the Materials and Methods, Procedure section (around line 135).

Also correct “except blood analysis” in line 275, which now sounds as if there was a difference between the 1st and 2nd pre-measurements regarding blood analysis, yet only one pre-measurement was performed for blood analysis.

Finally, add in 3.2.1. section that the 2nd pre-measurement is included in the statistical analysis.

  • Cortisol – line 198: cortisol has a marked circadian rhythm, blood must be drawn in the same phase (hour) of the day (preferably in the morning) to make the results comparable to each other, and there is a very significant difference between pre and post. Yet there is no information on the timing of the blood draw (nor on the timing of any other measurements).

CRP – line 199: CRP can be related to many conditions apart from chronic stress, especially to acute diseases. Were these also excluded when the measurements were done?

  • The description of the Intervention is incomplete. Explain in more details, the duration, the daily schedule, arrival, leaving, timing of measurements, how much time was spent in the forest actively (walking exercises), and with other program elements, lodging etc.
  • Finally, the lack of the follow-up of the blood composition (cathepsin-B, cortisol, IGF-1, CRP) is also a limitation of the study.

The study is otherwise appropriate, but the presentation of the results (and study design) has to be improved.

Author Response

A complex and prudent study, I liked the study design with a 1st and then a 2nd pre-measurement for verifying the chronic state, and then a post-intervention measurement and a 1-month follow-up, which are reasonable and valuable. The various aspects investigated simultaneously are also a strength of the work. Yet I have some questions and remarks, and there are a couple of points to clarify and explain more deeply in the article before finalizing it, especially regarding the pre-measurements and the description of the intervention.

  • All authors are deeply grateful for the good reviews of the reviewers. Although this study is a study design without the control group, it is also one of our objectives to propose an experimental study that could replace the control group by confirming a chronic condition. Once again we deeply appreciate your evaluation.

  1. I did not check it in details but references [10] and [12] may be mixed up.
  • Response:
  • We agree with you. The order of the references we presented is wrong, and we have corrected it appropriately. Thanks again for your meticulous comments.

  1. The authors wished to investigate the “elderly”, and in reference [1] (and in general as well) the elderly population is aged 65 years and over. Still, “60 years or older” participants were recruited with a mean age 61.90 +/- 1.14 years, who are younger. Why?
  • Response:
  • Your point is very pertinent, and the age of the subjects in our initial study was 65 years or older. As you are well aware, the classification of older people is generally as follows:
    • youngest-old (ages 65 and 74 years), middle-old (ages 75 and 84 years), and oldest-old (over 85 years)

https://www.ncbi.nlm.nih.gov/pmc/articles/PMC6301865/

  • However, the government of the Republic of Korea has announced that it cannot be selected for the study because the elderly 65 years and older are very vulnerable to COVID-19. So we had no choice but to relax the standards for the elderly. Nancy M. Petry considered people 55 and older to be older, and we chose those 60 and older to bring it closer to the standard of older people.

https://academic.oup.com/gerontologist/article/42/1/92/641498

  • We will make this more clear when COVID-19 enters the stabilization phase. I hope you understand the situation in Korea.

  1. The exclusion criteria are thorough and well, especially with excluding persons with regular (weekly) forest visits. Still, the last point in line 100, is difficult to interpret: “The exclusion criteria were as follows: … and (7) should perform moderate-intensity aerobic physical activities for more than 300 min.”
  • Response:
  • As described in P13 line 402, the WHO reported that health benefits appeared when the moderate-intensity aerobic exercise of 300 minutes or more per week was practiced in the elderly population. However, according to a survey by the National Statistical Office of South Korea reported in 2019, 67% of the elderly population aged 65 and over-performed moderate-intensity aerobic exercise for less than 150 minutes per week [1]. Therefore, in this study, the exclusion condition in “(7) should perform moderate-intensity aerobic physical activities for more than 300 min” was set for the sample group to represent the population.

  1. The performance and timing of a first AND second pre-measurement are quite obscure throughout the article until explanation in the Results section, in lines 272 and 273. Please explain this (why and when the 2 measurements were accomplished, and how close the 2nd pre-measurement was to the intervention) already in the Materials and Methods, Procedure section (around line 135).
  • Response:
  • All authors agree with the reviewers' good points. We have additionally described the following to clearly present information on experimental measurements.
  • P4 line 144-9: The participants performed two pre-measurements before participating in the program. To prove that a participant’s condition was chronic, the second pre-measurement was performed at an interval of at least one month after the first pre-measurement. On the day of the second pre-measurement, all participants underwent a forest healing anti-aging program for two days and one night over three weeks.

  1. Also correct “except blood analysis” in line 275, which now sounds as if there was a difference between the 1st and 2nd pre-measurements regarding blood analysis, yet only one pre-measurement was performed for blood analysis.
  • Response:
  • Thanks for the good advice. The sentence was ambiguous, so we edited it to make the meaning clearer as follow:
  • P9 line 296-7: Only blood analysis was performed at the second pre-measurement.

  1. Finally, add in 3.2.1. section that the 2nd pre-measurement is included in the statistical analysis.
  • Response:
  • Based on your suggestion, we additionally described the following in P9 line 291-4:
  • P9 line 294-7: Since there was no significant difference between the first and second pre-measurements, the second pre-measurement values were set as the pre-measurement value (Pre) to investigate the effect of the forest healing anti-aging program. Only blood analysis was performed at the second pre-measurement.

  1. Cortisol – line 198: cortisol has a marked circadian rhythm, blood must be drawn in the same phase (hour) of the day (preferably in the morning) to make the results comparable to each other, and there is a very significant difference between pre and post. Yet there is no information on the timing of the blood draw (nor on the timing of any other measurements).
  • Response:
  • You have made a very appropriate point, and we have taken this into account when performing our measurements. As you are well known, cortisol is a variable that is sensitive to circadian rhythms (biorhythm-sensitive) as well as meal consumption. Therefore, to exclude this, we collected blood at 8 am for the second pre- and post-measurements. This is additionally presented in the manuscript as follows:
  • P5 line 210-1: Blood was collected at 8 am for the second pre-measurement and post-measurement by direct injection by medical personnel with more than 10 years of experience.

  1. CRP – line 199: CRP can be related to many conditions apart from chronic stress, especially to acute diseases. Were these also excluded when the measurements were done?
  • Response:
  • As we pointed out, CRP is associated not only with chronic stress, but also with various physiological phenomena, including sleep disturbance, smoking, and the family's socioeconomic status [2-6]. Unfortunately, we did not take all of these into account when interpreting CRP. Based on the suggestion of a psychiatrist, we selected CRP as a biomarker to evaluate chronic stress. No subjects in this study had any other musculoskeletal disorders, chronic inflammatory diseases, neurological problems, or psychiatric disorders. Also, none of the participants were smokers, and drinking was strictly prohibited from immediately before the second pre-measurement to the post-measurement.

  1. The description of the Intervention is incomplete. Explain in more details, the duration, the daily schedule, arrival, leaving, timing of measurements, how much time was spent in the forest actively (walking exercises), and with other program elements, lodging etc.
  • Response:
  • We acknowledge that the information on the interventions used in this study has not been sufficiently described. According to your suggestion, we added the program schedule (Table 1, P6 line 260) and additional content (P8 line 261-4).
  • P6 line 263: Table 1
  • P8 line 264-7: During the intervention period of two days and one night, the participants stayed only at the facility within the National Center for Forest Therapy, and lodging and meals were also used within the facility. To move between programs and programs, walking was used.

  1. Finally, the lack of the follow-up of the blood composition (cathepsin-B, cortisol, IGF-1, CRP) is also a limitation of the study.
  • Response:
  • We agree with your good point. We were aware of this problem and attempted a follow-up measurement. However, the participants were very sensitive to repeated blood collection. In particular, due to COVID-19, there was a strong reluctance to direct contact and especially to blood collection, and unlike other measurements, we were not able to collect blood. We have additionally described the following as limitations. Once again we would like to thank you very much for your kind comments.
  • P14 line 451-4: Fourth, blood analysis was performed only in the second pre-measurement and post-measurement. Therefore, it cannot be confirmed in this study whether the changes in blood analysis persist until one month later.

The study is otherwise appropriate, but the presentation of the results (and study design) has to be improved.

  • In consideration of your good points, we have revised many contents of the main manuscript. We hope this will satisfy you. All authors once again thank you very much for your good points.

Reviewer 3 Report

This is a very interesting intervention and offers a realistic therapeutic option.

Throughout the authors use the word ‘elderly’ – would suggest a change to ‘older people’ as this is more acceptable to older people worldwide, and certainly amongst Western readership the word ‘elderly’ is considered stigmatised.

P2 line 47: phrase “it’s cognitive impairment” is not clear – please reword.

P2 line 56: MCI is not explicitly a pre-dementia stage. The concept is not really a diagnostic entity in its own right due to numerous heterogenous definitions and instability over time (many meeting criteria for MCI revert to criteria for normal cognition within two years.

P2 line 63: curative interventions are not currently available and unlikely that the present study represents a curative intervention, so this language should be changed.

P4 line 142: not clear at all when blood was taken as it various says different time points throughout the paper. Suggest clarification.

P5 line 209: Not clear what these values relate to – are they the psychometric properties of this measure?

Measures generally need to state the scoring range for all.

Table 1: Good to see contents of programme but how do these relate to each session? It would be good to include an example session plan and more details of how these were delivered e.g. how many participants did each session in one go (was it all of the participants together or in smaller groups?), how many facilitators, what was the timeframe, any other important info like transport, support for carers, subsistence and accommodation.

P10 line 329: You should state here that there were improved scores on measures of cognitive ability as you can’t definitely say that it improved cognitive function itself – can’t discount the potential for practice effects or feeling less stressed due to the intervention might have increased performance.

P10 line 332 – this is too much of a leap to suggest and it is more likely that forest healing has benefitted both positively with the potential for an interaction should the intervention be practised for a longer time frame.

P11 line 338: It is a leap to say that insufficient relationships improved because you can’t definitively state that their relationships were insufficient from the baseline data – there are only three questions in the WHOQOL_BREF that make up the social relationships domain and that’s not enough evidence to decide if their relationships were insufficient or not. Further, improving by one point might be statistically significant but it’s not particularly meaningful clinically/in the real world.

P11 line 340: You can’t show that they obtained emotional stability because you are not able to demonstrate from your baseline data that participants were emotionally unstable. The scores might be higher than normative values in a general population, but it’s not clear that they represent severe mood disorder – although clarification in the range of scores would assist this interpretation but my understanding is that a maximum score of 200 is possible which is far greater than the scores presented in this paper.

P11 line 362: Not sure what this point adds

P11 line 364: “blood muscle and stress factors” is unclear, suggest reword.

Author Response

This is a very interesting intervention and offers a realistic therapeutic option.

  1. Throughout the authors use the word ‘elderly’ – would suggest a change to ‘older people’ as this is more acceptable to older people worldwide, and certainly amongst Western readership the word ‘elderly’ is considered stigmatised.
  • Response:
  • We appreciate your attentive comments. As per your point, we have changed 'elderly' to 'older people' throughout the manuscript.

  1. P2 line 47: phrase “it’s cognitive impairment” is not clear – please reword.
  • Response:
  • Based on your good point, we have revised the sentence as a whole. If the changes we made didn't work for you, please point them out again.
    • P2 line 48-52: Dementia is one of the main causes of disability and dependence in older people around the world. Cognitive function of older people with dementia is significantly lower than that of normal older people of the same age, which negatively affects overall brain activities such as memory, thinking, understanding, learning, language judgment, and calculation.

  1. P2 line 56: MCI is not explicitly a pre-dementia stage. The concept is not really a diagnostic entity in its own right due to numerous heterogenous definitions and instability over time (many meeting criteria for MCI revert to criteria for normal cognition within two years.
  • Response:
  • All authors agree with your good point. Based on your suggestion, we have revised that sentence as follow:
    • P2 line 55-7: MCI refers to an intermediate stage between the decline in expected cognitive function due to normal aging and severe cognitive decline due to dementia [3].

  1. P2 line 63: curative interventions are not currently available and unlikely that the present study represents a curative intervention, so this language should be changed.
  • Response:
  • We agree with your good point. Considering that curative intervention for improving symptomatic intervention in MCI is currently unclear, we deleted the word.

  1. P4 line 142: not clear at all when blood was taken as it various says different time points throughout the paper. Suggest clarification.
  • Response:
  • Based on your appropriate comments, we have revised the sentence as follow:
  • P4 line 157-60: Except for blood analysis, all variables were evaluated at the first and second pre-measurement, post-measurement 4 hours after the last intervention, and one-month follow-up-measurement. Blood collection and analysis were performed at the second pre-measurement and post-measurement.

  1. P5 line 209: Not clear what these values relate to – are they the psychometric properties of this measure? Measures generally need to state the scoring range for all.
  • Response:
  • Thanks for the good comments. The value of p5 line 210 is the test-retest reliability of the senior fitness test for the elderly with cognitive impairment. However, we found that the original sentence was ambiguous, and references were missing. We have corrected the sentence and provided an appropriate reference.
  • P6 line 225-7: When the SFT was performed on older people with cognitive impairment, the values of test-retest reliability ranged from 0.93 to 0.98 [30].

  1. Table 1: Good to see contents of programme but how do these relate to each session? It would be good to include an example session plan and more details of how these were delivered e.g. how many participants did each session in one go (was it all of the participants together or in smaller groups?), how many facilitators, what was the timeframe, any other important info like transport, support for carers, subsistence and accommodation.
  • Response:
  • In consideration of your suggestion, we have presented the program schedule as a table, and the content is additionally described in the text.
    • P6 line 263: Table 1
    • P8 line 264-7: During the intervention period of two days and one night, the participants stayed only at the facility within the National Center for Forest Therapy, and lodging and meals were also used within the facility. To move between programs and programs, walking was used.

  1. P10 line 329: You should state here that there were improved scores on measures of cognitive ability as you can’t definitely say that it improved cognitive function itself – can’t discount the potential for practice effects or feeling less stressed due to the intervention might have increased performance.
  • Response:
  • We agree with your comments, and we have revised the description as follows:
  • P12 line 349-51: Similarly, the forest healing program improved scores on measures of cognitive ability of older people with mild cognitive impairment by 5.1%.

  1. P10 line 332 – this is too much of a leap to suggest and it is more likely that forest healing has benefitted both positively with the potential for an interaction should the intervention be practised for a longer time frame.
  • Response:
  • We acknowledge that there was a leap in that expression. We have modified the expression based on your advice.
  • P12 line 351-3: Also, as the scores of BDI and DD of POMS decreased significantly, it was found that the forest healing anti-aging program was effective in improving depression in older people.

  1. P11 line 338: It is a leap to say that insufficient relationships improved because you can’t definitively state that their relationships were insufficient from the baseline data – there are only three questions in the WHOQOL_BREF that make up the social relationships domain and that’s not enough evidence to decide if their relationships were insufficient or not. Further, improving by one point might be statistically significant but it’s not particularly meaningful clinically/in the real world.
  • Response:
  • First of all, all authors would like to thank you very much for your good point. In general, although there is a negative correlation that the social relationship decreases with age [1], such a trend cannot be seen as clear in this study. Therefore, the 'insufficient' mentioned was deleted.
  • As you pointed out, although there were statistically significant changes in the results of this study, there may be no clinical significance due to small differences in scores. For this reason, various studies are suggesting criteria such as clinical mean difference when observing symptom changes caused by various diseases. However, MCI is not a patient and it is difficult to show significant improvement because the baseline value is very high. Therefore, we interpret a small change in a score as significant.
  • Also, we had fewer participants due to COVID19. This will be reinforced in future studies, and we will use other cognitive evaluation methods to present better results.
  • Thanks again for the good point.

  1. P11 line 340: You can’t show that they obtained emotional stability because you are not able to demonstrate from your baseline data that participants were emotionally unstable. The scores might be higher than normative values in a general population, but it’s not clear that they represent severe mood disorder – although clarification in the range of scores would assist this interpretation but my understanding is that a maximum score of 200 is possible which is far greater than the scores presented in this paper.
  • Response:
  • Numerous studies have been used POMS to evaluate the severity of mood disorders, and this tool shows high reliability and validity. However, as you suggested, the tool can calculate the ‘total mood disturbance’ score ranging from -32 to 200, and does not provide a quantitative reference score for severe mood disturbance.
  • However, since this tool calculates the ‘total mood disturbance’ score by subtracting positive emotional scores from negative emotional scores, it can be said that the higher the ‘total mood disturbance’ score, the more negative emotions are felt. In this study, the scores tended to decrease significantly, suggesting that the participants' mood states improved, which is consistent with many previous studies [2-6].

  1. P11 line 362: Not sure what this point adds
  • Response:
  • We appreciate your advice. We determined that the sentence was unnecessary for the content of the text and deleted it.

  1. P11 line 364: “blood muscle and stress factors” is unclear, suggest reword.
  • Response:
  • All authors would like to thank you for your good comments. We changed 'blood muscle and stress factors' to 'in factors related to muscle and stress in the blood'.
  • P12 line 382-4: In factors related to muscle and stress in the blood, cortisol, a stress hormone, was significantly reduced, indicating that the forest healing program was effective in reducing the stress level of the participants.
